# Psychological Factors, Digital Health Technologies, and Best Asthma Management as Three Fundamental Components in Modern Care: A Narrative Review

Pasquale Caponnetto [1,2], Graziella Chiara Prezzavento [1,*], Mirko Casu [1,3], Riccardo Polosa [2,4] and Maria Catena Quattropani [1]

1   Department of Educational Sciences, Section of Psychology, University of Catania, 95123 Catania, Italy; pcapon@unict.it (P.C.); mirko.casu@phd.unict.it (M.C.); maria.quattropani@unict.it (M.C.Q.)
2   Center of Excellence for the Acceleration of Harm Reduction (CoEHAR), University of Catania, 95121 Catania, Italy; polosa@unict.it
3   Department of Mathematics and Computer Science, University of Catania, Viale Andrea Doria 6, 95125 Catania, Italy
4   Department of Clinical & Experimental Medicine, University of Catania, 95125 Catania, Italy
*   Correspondence: graziellachiara.prezzavento@studium.unict.it

**Abstract:** New digital interventions have shown potential in managing asthma and improving patients' quality of life compared with conventional interventions. Our objective was to conduct an exhaustive survey of the application of digital health technologies in evaluating, treating, and self-managing psychological and psychopathological elements linked to asthma. We analyzed a compendium of research papers pertaining to asthma, encompassing themes such as outdoor air pollution, early life wheezing disorders, atopic dermatitis, digital strategies for asthma self-management, psychiatric conditions and asthma, familial impacts on pediatric asthma, and the utilization of mobile health apps for managing asthma. We scrutinized six chosen studies to evaluate the capacity of digital health technologies to enhance the management and treatment outcomes of psychological factors related to asthma. The studies under review indicate that eHealth interventions, mixed reality instruments, mHealth technology-augmented nurse-led interventions, and smartphone apps incorporating Bluetooth-enabled sensors for asthma inhalers can markedly enhance self-management of symptoms, quality of life, and mental health outcomes, particularly in children and adolescents with asthma. Nonetheless, additional research is required to ascertain their efficacy and practicability across diverse populations and environments. Customized interventions that cater to the unique needs and preferences of asthma patients and associated psychological elements are vital to ensure the persistent and equitable adoption of these technologies. We underscore the significance of addressing psychological elements in the management and treatment of asthma and advocate for ongoing research and development in this domain.

**Keywords:** asthma; eHealth; mobile apps; digital health interventions; mixed reality; depression; anxiety; clinical psychology; narrative review





## 1. Introduction

Asthma, a chronic respiratory ailment that afflicts both adults and children, is characterized by inflammation and constriction of the airways, resulting in difficulty in breathing. The World Health Organization (WHO) has highlighted that asthma is often inadequately diagnosed or managed, particularly in low- to middle-income nations [1]. Globally, the disease affects an estimated 100 to 150 million individuals, with the annual death toll associated with the condition, as per WHO data, standing at approximately 180,000 [2]. In Italy, the prevalence of asthma is around 10% among children, with severe asthma affecting 2% of asthmatic children and adolescents [3,4]. Among the adult population in

Italy, the prevalence of asthma is approximately 8.4% [5]. The primary causes and risk factors for asthma encompass genetic predispositions [6], environmental influences [7], respiratory infections [8], and atopy [9]. The emergence of novel technologies has facilitated the introduction of efficacious treatment methodologies in clinical environments, including SMS-based interventions [10], smartphone applications [11], virtual and augmented reality [12,13], and commercial video games [14]. These are encompassed within the categories of electronic Health (eHealth) and mobile Health (mHealth). eHealth involves the utilization of information and communication technologies for health, such as electronic health records, personal health records, and clinical decision support tools. It encompasses a broad range of applications aimed at improving healthcare delivery, management, and outcomes through the integration of digital technologies into various aspects of healthcare systems. Examples of eHealth initiatives include telemedicine, telehealth, health information exchange systems, and health information portals. On the other hand, mHealth pertains to medical and public health practices supported by mobile devices, such as mobile phones, patient monitoring devices, personal digital assistants, and other wireless devices. mHealth technologies leverage the ubiquitous nature of mobile devices to deliver healthcare services and information remotely, allowing for real-time monitoring, communication, and intervention. This includes a wide array of applications, such as mobile health apps, wearable devices, text messaging programs, and remote monitoring tools, which are designed to empower individuals to manage their health and wellness effectively. Together, eHealth and mHealth play a pivotal role in transforming healthcare delivery and improving patient outcomes by leveraging the capabilities of digital technologies to enhance access, efficiency, and quality of care. By harnessing the power of information and communication technologies, healthcare providers can deliver more personalized, timely, and effective interventions, while patients can actively engage in self-management and decision-making processes to achieve better health outcomes [15]. In the context of asthma, these novel digital interventions aim to enhance asthma management, self-monitoring, and adherence to treatment regimens. They have been positively received by patients, who generally value the usability of these tools and express a willingness to continue treatments [16–18], thereby improving adherence. Specifically, multifunctional mobile health applications (mHealth apps) have demonstrated potential in managing asthma and improving patients' quality of life compared to conventional interventions [19]; additionally, digital inhaler health platforms, which employ digital inhalers to monitor dosage timing and dates, have proven effective in managing both asthma and medication [20]. These platforms have fostered a collaborative care environment between healthcare providers and patients and provide a more comprehensive understanding of actual inhaler usage, demonstrating potential, with evidence indicating positive effects on various outcomes, including knowledge, activity limitations, self-care, quality of life, and medication usage [21]. The development of digital interventions is often underpinned by psychological models and theories aimed at enhancing their impact and utilization, guiding their optimal design and delivery [22]. In fact, theory-based digital interventions can improve asthma self-management outcomes, such as medication adherence, asthma control, and quality of life [23].

Within the realm of psychology, it is recognized that asthma can be influenced by a variety of psychological and psychopathological factors: individuals afflicted with asthma frequently exhibit comorbidities, demonstrating a higher incidence of anxiety and depressive disorders compared to the general populace [24]. These psychological elements can adversely affect asthma control, adherence to treatment protocols, and overall quality of life [25]. Stress has the potential to intensify asthma symptoms, and interventions aimed at managing stress may contribute to improved asthma outcomes [26]. Furthermore, the relationship between asthma and psychological factors can be bidirectional: while psychological factors can influence asthma symptoms, the symptoms of asthma can also precipitate increased psychological distress [27]. Parental psychological factors can also exert an impact on the outcomes of children's asthma. Specifically, parental anxiety and depression have been linked to poorer asthma control in children [28]. Collectively, these

studies underscore the intricate interplay between psychological and psychopathological factors and asthma, highlighting the necessity of addressing these factors in the management and treatment of asthma. Table 1 provides a summary of the paper's contribution to the extant literature on digital health interventions for managing psychological factors related to asthma.

**Table 1.** The table delineates the identified problem or issue, summarizes the existing knowledge on the subject, and articulates the novel contribution of the paper to the extant literature.

| Problem or Issue | What Is Already Known | What This Paper Adds |
| --- | --- | --- |
| Addressing psychological aspects linked to asthma through the use of digital health technologies. | Prior studies have highlighted the potential benefits of eHealth approaches, mixed reality tools, and mobile applications in improving symptom management, quality of life, and mental health outcomes. | Digital health technologies have been identified to significantly boost symptom self-management, enhance quality of life, and positively influence mental health outcomes, particularly in children and adolescents with asthma. The analysis of selected studies indicates the potential efficacy of these digital interventions in tackling psychological factors in the management and treatment outcomes of asthma. |

The objective of our literature review is to explore the current advancements in digital health interventions, specifically pertaining to the evaluation and treatment of psychological aspects associated with asthma, such as stress, anxiety, depression, and coping mechanisms employed to mitigate the emotional distress induced by asthma. Additionally, we assess the feasibility of utilizing these cutting-edge technological tools for conducting observational clinical studies. We also examine the effectiveness and deployment of digital aids for self-management for both adult and pediatric asthma patients.

## 2. Materials and Methods

### 2.1. Research Guidelines

This review was undertaken in accordance with the Preferred Reporting Items for Systematic Reviews and Meta-Analyses (PRISMA) for systematic reviews [29].

### 2.2. Research Question

The guiding research question for this review was as follows: "How do digital technologies for self-management and treatment (such as mobile apps, virtual reality, and wearable technology) compare to traditional asthma management methods in terms of psychological factors (stress, anxiety, depression) and their impact on asthma in adults and youth?" This question was framed using the patient, intervention, comparison, outcome (PICO) framework [30] to guide the exploration of the effectiveness of digital technologies in addressing psychological and psychopathological factors associated with asthma in both adult and youth populations. Each article included met the following criteria: the study population comprised adults and youth with asthma of any sex and socioeconomic status; the intervention involved digital technologies for self-management or treatment of asthma-related psychological and psychopathological factors (mobile apps, virtual reality, and wearable technology); some research compared traditional asthma management methods with those incorporating digital technologies; and the outcome evaluated the effectiveness of digital technologies for self-management and/or treatment of psychological and psychopathological factors related to asthma.

### 2.3. Literature Search

From March to June 2023, we conducted several searches on our internal search engine Summon [31] using different search terms combinations. We analyzed articles from MEDLINE (PubMed, PubMed Central, Health & Medical Collection, and Web of Science databases) and EMBASE (Elsevier), focusing on results that discussed the relationship

between asthma and digital technologies, such as mobile applications, virtual reality, and wearable devices, and their correlation with psychological factors like stress, anxiety, and depression in children, adolescents, and young adults. Regarding EMBASE, we did not find any results using the search strings.

The first search was conducted using the terms "(asthma) AND ((smartphone applications) OR (mobile apps)) AND ((psychological factors) OR (psychopathological factors))". Out of the 12 papers found, 9 were duplicates and 2 were irrelevant, leaving 1 paper for further analysis.

The second search used the terms "(asthma) AND ((smartphone applications) OR (mobile apps)) AND ((stress) OR (anxiety) OR (depression))". Out of the 99 papers found, 68 were duplicates, leaving 3 papers for further analysis.

The final search used the terms "(asthma) AND ((Wearable technology) OR (Telemedicine)) AND ((stress) OR (anxiety) OR (depression))". Out of the 164 papers found, 104 were duplicates, leaving 2 papers for further analysis.

All the studies published since 2015 were included. There were no limitations based on language. The reference lists of the included studies underwent additional scrutiny to identify additional potential studies. Two authors of the review independently examined and chose studies from the conducted searches.

### 2.4. Eligibility Criteria

We included every article that met the following criteria:

(a)  All studies, qualitative protocols, participatory studies, and original studies that were published in indexed journals and listed in PubMed, PubMed Central, Health & Medical Collection, and Web of Science databases.
(b)  Studies about eHealth and mHealth interventions associated with asthma's psychological and psychopathological factors, such as stress, anxiety, and depression in children, adolescents, and young adults.
(c)  Studies published from 2015 to the present.

### 2.5. Exclusion Criteria

We excluded every article meeting the following criteria:

(a)  Studies not related to digital technologies associated with asthma's psychological and psychopathological factors, such as stress, anxiety, and depression.
(b)  Studies that focus on the clinical management of asthma, such as pharmacological interventions.
(c)  Studies that do not examine the psychological factors associated with the use of digital technologies for asthma.
(d)  Studies published in non-scientific or non-peer-reviewed sources.
(e)  Studies published prior to 2015.

### 2.6. Data Collection

The data collection process was independently carried out by two reviewers. Any discrepancies that arose were addressed through dialogue or with the aid of a third reviewer.

## 3. Results

### 3.1. Characteristics of the Included Studies

The search process resulted in a total of six full-text articles, following which ten full-text articles were evaluated for eligibility. The article selection procedure, as reported in the PRISMA and compiled in line with the guidelines for updates of systematic reviews, is depicted in Figure 1. The primary characteristics of the included studies are elucidated below.

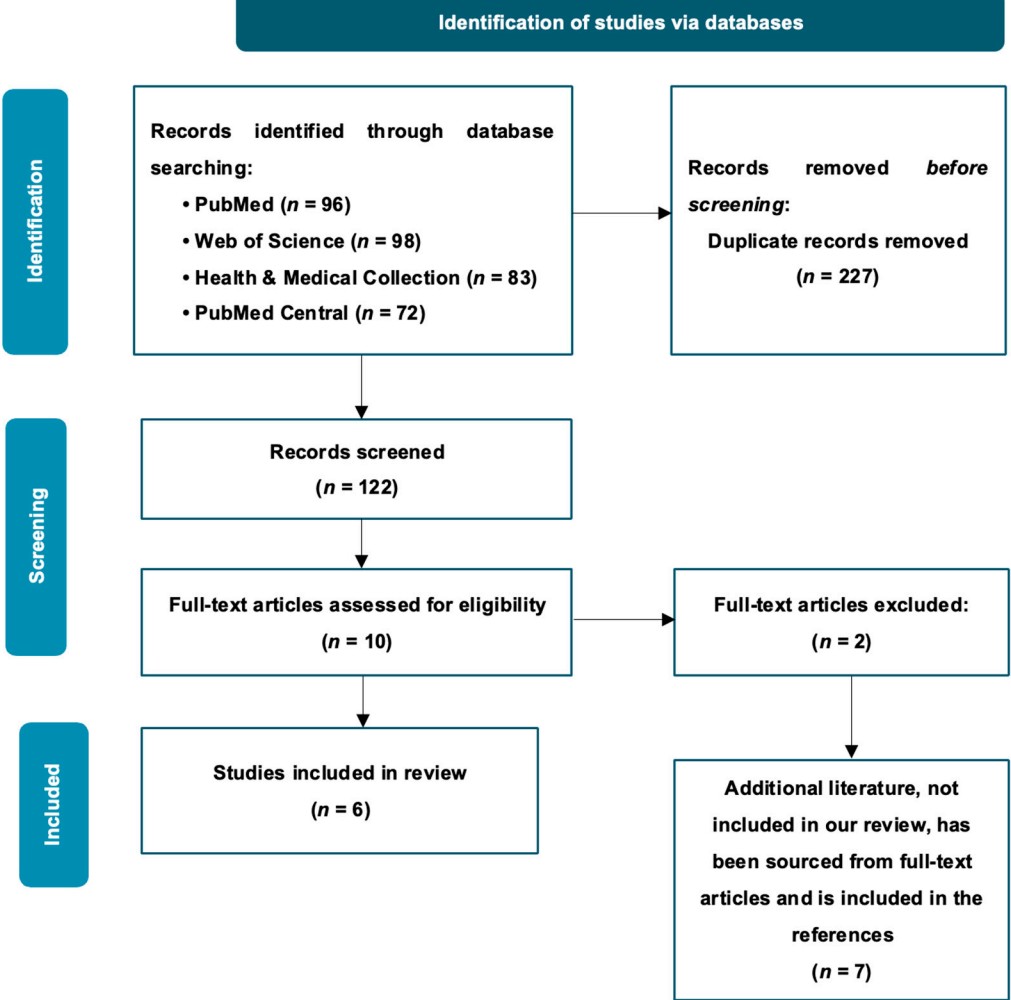

**Figure 1.** The PRISMA (Preferred Reporting Items for Systematic Reviews and Meta-Analyses) flow diagram [29].

The articles encompassed in this review delve into the utilization of eHealth interventions and digital health technologies for the assessment, treatment, and self-management of psychological and psychopathological facets linked to asthma in children, adolescents, and young adults. The review includes qualitative protocols, participatory studies, and articles. The studies were subjected to qualitative analysis, and the findings were amalgamated to furnish an overview of the contemporary state of research in this domain. A synopsis of these studies is presented in Table 2. The constraints of the studies and the ramifications for prospective research are also deliberated upon.

**Table 2.** Summary of studies included in the results section.

| Authors | Year | Type of Intervention/Tool | Type of Paper | Outcome |
|---|---|---|---|---|
| Nichols et al. [32] | 2020 | mHealth intervention | Study Protocol | The efficacy of the mHealth strategy (MATADORS) in aiding symptom self-regulation among young individuals with asthma. |
| Peters et al. [33] | 2017 | Preferences for an asthma self-management app | Participatory Study | Determining the psychological requirements and preferences of young individuals for an asthma self-regulation application. |

**Table 2.** *Cont.*

| Authors | Year | Type of Intervention/Tool | Type of Paper | Outcome |
|---|---|---|---|---|
| Neves et al. [34] | 2021 | Health and fitness mobile apps | Secondary analysis of observational studies | Factors influencing the utilization of health and fitness mobile applications by individuals with asthma. |
| Dzubur et al. [35] | 2015 | Smartphone application that integrates sensors for asthma inhalers | A letter to the editor | The application can facilitate comprehension of real-time experiences of adolescent asthma patients, enhance adherence to treatment, customize therapies, and improve communication between patients and providers. |
| Shah et al. [36] | 2022 | Digital Health Interventions | Scoping review | DHIs are predominantly utilized for treatment, with a primary emphasis on depression, and are typically disseminated through web-based platforms. Nonetheless, their integration into routine care is restricted. |
| Sharrad et al. [37] | 2021 | Mixed reality technology | Qualitative protocol | The manuscript presents a project aimed at creating and evaluating mixed reality instruments for administering cognitive and behavioral therapies to young individuals with asthma. The outcomes of the project are currently unavailable. |

### 3.2. eHealth and mHealth Strategies: Tackling Psychological Elements in Children and Adolescents with Asthma and Persistent Health Conditions

In this segment, we will scrutinize articles that delve into the utilization of eHealth and mHealth tools for the treatment, management, or self-monitoring of psychological factors associated with young individuals, ranging from children to adolescents.

The inaugural study we examine is the one spearheaded by Nichols et al. [32] who have delineated the development and feasibility testing of a mHealth intervention, termed MATADORS, aimed at enhancing symptom self-management among youth (ages 10–17) with asthma and obesity. The intervention amalgamates motivational enhancement and behavioral activation strategies within a family-centric model and an innovative mobile app that encompasses educational content, symptom monitoring, activity tracking, and photo/video/voice diaries. The authors reported the outcomes of a pilot study that randomized 30 youth with asthma and obesity to either MATADORS or a control condition. The intervention group partook in a six-week nurse-guided mHealth program, while the control group received standard care. The feasibility outcomes encompassed recruitment, reach, adherence, satisfaction, and participant burden. The preliminary outcomes included fatigue, pain, self-efficacy, anxiety, sleep, depression, and quality of life, which were measured at baseline, six, and twelve weeks. The authors concluded that MATADORS is a feasible and acceptable intervention with the potential to enhance symptom self-management and quality of life for youth with asthma and obesity. However, they did not report the statistical significance of the preliminary outcomes, rendering it unclear whether MATADORS was effective in terms of reducing these symptoms and improving these outcomes.

Subsequently, Peters and colleagues [33] conducted a participatory study to comprehend the experiences, needs, and ideas of young individuals with asthma. The objective was to identify the prerequisites for an asthma app that would be engaging and effective in augmenting their well-being. The study comprised 20 participants aged between 15 and 24 years who completed a workbook and participated in a workshop, culminating in the creation of 102 participant-generated artifacts. The data underscored that psychological aspects such as anxiety and impediments to autonomy, competence, and relatedness were perceived by young individuals with asthma as significant influences on their quality of life. The study proposes that an app designed for young individuals with asthma should provide support for mental health factors associated with lived experience, in conjunction

with practical features for managing asthma. These findings could potentially influence the design of technologies intended to assist individuals with chronic illnesses more broadly.

### 3.3. mHealth Strategies: Factors Influencing the Use of Health and Fitness Applications by Individuals with Asthma

Fitness applications can exert a significant influence on psychological health by offering several key advantages [38]. They provide motivation and support, assisting users in setting objectives, tracking progress, and receiving feedback and encouragement for their physical activity. This can result in an enhancement in self-esteem and a sense of achievement. Fitness applications also foster engagement and enjoyment by aiding users in selecting the most effective and enjoyable forms of physical activity tailored to their needs, thereby promoting increased engagement and a more positive relationship with exercise. Moreover, the convenience and accessibility of fitness applications facilitate the incorporation of regular physical activity into users' lives, which can ameliorate mood and alleviate symptoms of stress, anxiety, and depression. However, it is crucial to note that while fitness applications can contribute to psychological well-being, they do not serve as a substitute for professional mental health care [38]. In this context, Neves and colleagues [34] conducted a study to assess the usage of general health and fitness applications among individuals with asthma and to identify the factors that influence their usage. The majority of the participants reported having good overall health. However, the incidence of anxiety and depression was found to be 34.3% and 11.9%, respectively. Approximately 41.1% of the participants reported using health and fitness mobile applications. The research revealed that individuals who were single and those with more than 10 years of education were more predisposed to use these applications. Furthermore, participants with higher digital literacy scores were more likely to use the applications. Interestingly, participants exhibiting symptoms of depression were less likely to use health and fitness applications. The study underscores the importance of understanding the barriers and motivators for application usage among patients with lower education levels, lower digital literacy, or depressive symptoms. This understanding is crucial for devising tailored interventions that ensure sustained and equitable use of these technologies.

### 3.4. mHealth Strategies: Real-Time Monitoring and Management of Asthma Symptoms via a Smartphone Application Incorporating Inhaler Sensors

Dzubur et al. [35] expounded on the creation of a smartphone application that amalgamates Ecological Momentary Assessment (EMA) and Bluetooth-enabled sensors for asthma inhalers to monitor real-time asthma symptoms, social and physical context, behavior, stress, and inhaler usage. The efficacy of long-term asthma management is influenced by behavioral elements such as medication adherence and psychosocial stress. The application was devised by a multidisciplinary team and trialed on a small cohort of Hispanic middle and high school students. The app employed signal-contingent and event-contingent EMA sampling activated by asthma inhaler usage. It prompts users with electronic surveys at random intervals and post-inhaler usage, gathering data on various factors such as positive and negative affect, stress, energy, fatigue, activity type, social and physical contexts, and asthma symptoms. The app shows potential in assisting researchers and clinicians to attain a deeper comprehension of the real-time experiences of teenage asthma patients. It could bolster adherence to treatment regimens, customize treatments to cater to specific needs, and enhance communication between patients and providers. Future research should focus on augmenting adherence rates, expanding the findings to populations beyond non-Hispanics, and assessing the health literacy of adolescents.

### 3.5. Digital Health Strategies: Managing Depression and Anxiety in Asthma and Persistent Health Conditions

Digital health interventions (DHIs) are health initiatives delivered via digital technologies, such as web-based platforms, mobile applications, or telehealth systems [39]. DHIs may hold the potential to enhance the mental health of individuals with asthma,

who frequently encounter depression and anxiety as concurrent conditions [40]. In this context, Shah et al. [36] undertook a scoping review of the literature on DHIs for the prevention, detection, or treatment of depression and anxiety among individuals with chronic conditions, including asthma. The 53 studies in this review detailed 36 distinct DHIs for prevention (8%), detection (9%), and treatment (83%) of mental health issues in individuals with chronic conditions. The most prevalent technologies employed were web-based platforms (38%) and mobile devices (32%). The majority of studies concentrated on depression (68%), with some addressing both depression and anxiety (26%). DHIs typically encompassed education (57%), psychological therapy (53%), and mental health status monitoring (26%). Additional components included peer support (17%), communication with healthcare providers (17%), mindfulness (6%), and chat rooms or forums (13%). The majority of interventions were guided (60%), with guidance provided by various professionals, such as nurses, psychologists, and allied health professionals. Guidance served various functions, including answering queries, providing information, promoting engagement, and monitoring symptoms. The delivery methods for guidance varied, with some utilizing a combination of in-person, phone calls, and text messages. The study revealed that these tailored DHIs have the potential to significantly enhance the mental health outcomes of individuals with asthma, who frequently suffer from depression and anxiety as concurrent conditions. In particular, stepped-care models demonstrated promise for integrating DHIs into standard care, however, their implementation remains limited.

*3.6. Protocol for Mixed Reality Instruments: Delivering Cognitive Behavioral Therapy to Mitigate Symptoms of Severe Psychological Stress in Young People with Asthma*

A qualitative protocol, proposed by Sharrad and colleagues [37], was devised to explore the potential of employing mixed reality tools, encompassing augmented reality (AR), virtual reality (VR), and holographic technology, for administering cognitive behavioral therapy to manage symptoms of heightened psychological distress in young individuals with asthma. "Mixed reality" refers to a spectrum of technologies that blend elements of both the physical and digital worlds, allowing users to interact with both simultaneously. In the context of this study, mixed reality encompasses several specific technologies: augmented reality (AR) overlays digital information onto the real-world environment, typically viewed through a device such as a smartphone or smart glasses. Users can see virtual objects or information superimposed on the physical world; virtual reality (VR) creates immersive, computer-generated environments that users can interact with via specialized headsets or goggles. VR typically blocks out the physical world entirely, replacing it with a simulated digital environment; holographic technology creates three-dimensional images or projections that appear to exist in physical space, allowing users to interact with virtual objects as if they were real. The efficacy of this protocol is anchored in the utilization of mixed reality tools such as AR, VR, and holographic technology for delivering evidence-based cognitive behavioral therapy (CBT) to young individuals experiencing elevated psychological distress. Mixed reality technologies possess the ability to address low health literacy [41,42], can be tailored based on population characteristics, enhance engagement with content [43], broaden the geographic reach and accessibility of information [44], and facilitate real-time content updates. The project intends to employ a qualitative action research framework, moderator guides, and qualitative research to examine the feasibility and acceptability of using mixed reality tools for delivering CBT to manage symptoms of heightened psychological distress in young individuals with asthma. The project would also develop mixed reality-enabled CBT resources and evaluate their acceptability and usability through one-on-one interviews with young people, parents, and health professionals. The data would be analyzed through three pre-determined perspectives, and the mixed reality resources would be refined based on qualitative feedback. Upon completion of the project, the data would be compiled for use in publications and conferences, and reports would be disseminated to participants and relevant stakeholders. The study plans to involve 30 participants, comprising 10 young individuals with asthma, 10 parents of

these young individuals, and 10 healthcare professionals. The information gathered from this study would be utilized to refine the mixed reality resources for future feasibility studies, and the objective of the study is to mitigate the impact of asthma by improving access to evidence-based treatments for heightened psychological distress in young individuals with asthma.

## 4. Discussion

The conclusions of this review underscore the potential of digital health interventions in addressing psychological factors associated with asthma, particularly in children and adolescents. The studies included demonstrated that eHealth interventions, mixed reality tools, mHealth technology-enhanced nurse-guided interventions, and smartphone applications integrating EMA and Bluetooth-enabled sensors for asthma inhalers can significantly enhance symptom self-management, quality of life, and mental health outcomes in this demographic [32,35]. The systematic review by McGar et al. [45] sheds light on the potential benefits of eHealth interventions in addressing psychological sequelae associated with chronic medical conditions in pediatric populations. Their findings suggest that tailored eHealth interventions, encompassing self-management education and interactive asthma monitoring, have the potential to improve disease management and enhance the quality of life in children with persistent asthma. Moreover, the authors advocate for the integration of evidence-based theoretical frameworks, such as cognitive behavioral therapy or problem-solving therapy, into future eHealth interventions to meet the specific needs of children with medical conditions. In contrast, the systematic review conducted by Thabrew et al. [46] presents a more nuanced perspective on the efficacy of eHealth interventions for anxiety and depression in children and adolescents with long-term physical conditions, including asthma. Despite incorporating five trials involving various interventions, the authors found very low-quality evidence regarding the superiority and acceptability of eHealth interventions compared to standard treatments. This uncertainty underscores the need for further research to develop and evaluate effective and acceptable e-health interventions tailored to the unique challenges faced by children and adolescents with chronic medical conditions.

However, while the findings of these studies highlight the potential of eHealth interventions in augmenting disease management and addressing psychological distress in pediatric populations with asthma, it is crucial to note that the domain of digital health interventions for the treatment of anxiety or depression in conjunction with long-term physical conditions, including asthma, remains limited. Additional research is required to ascertain the effectiveness and feasibility of these interventions across diverse populations and settings [35]. Tailored interventions that cater to the specific needs and preferences of patients with asthma and associated psychological factors are vital for ensuring sustained and equitable utilization of these technologies [35]. Future studies should concentrate on enhancing methodological rigor, generalizability, and long-term outcomes, as well as addressing concerns about accessibility, privacy, and data security. The development of personalized interventions and the integration of digital health technologies into routine clinical care holds the potential to affect a significant difference in public health for individuals living with chronic conditions, including asthma. In addition, another critical aspect to consider when discussing digital health interventions is that, unfortunately, not everyone has equal access to the use of digital health technologies among different demographic groups based on socioeconomic status, geographic location, education level, and age on the adoption and effectiveness of eHealth and mHealth interventions for asthma management [47]. It is clear that individuals from underserved communities or those with limited access to technology may face significant barriers to benefiting from digital health interventions. Factors such as lack of Internet connectivity, low digital literacy, and limited access to smartphones or computers can hinder their ability to use these technologies effectively. Addressing the digital divide is critical to ensuring equitable access to health resources and interventions. By recognizing and proactively addressing the digital

divide, we can take significant steps toward achieving health equity and ensuring that all individuals, regardless of socioeconomic background, have an equal opportunity to benefit from advances in digital health technologies.

As authors, we have been engaged in the psychological and psychotherapeutic field for years, and we are well aware of how psychological and psychopathological factors such as anxiety and depression, for instance, can diminish people's quality of life. The increasingly comprehensive use of new technologies opens numerous avenues, and the creation of treatment programs for such factors when they coexist with asthma is certainly an unfortunately under-explored field, but one of substantial social and health impact. The presence of few studies in this review is indicative of just that: we are aware that asthma, like other chronic conditions, causes discomfort in people, discomfort that can evolve into genuine psychopathological disorders. Our objective with this review is therefore at least to pose the question to the scientific community, highlighting what is available for consultation at the time of writing. We hope that this will stimulate increased interest and development, as well as the application, of the new technologies available in digital health to mitigate the impact that negative psychological factors can have on people with asthma.

*Limitations*

The articles reviewed offer insights into the application of digital health interventions for the management of asthma and associated psychological factors, yet there are existing limitations. The breadth of the review is confined to a particular set of studies, owing to the paucity of studies in the literature specifically addressing digital health interventions for managing asthma and related psychological factors. The efficacy of these interventions may exhibit variability across different populations and settings, and the need for their personalization warrants further exploration. This review does not tackle the technological hurdles in adopting these interventions, nor does it contemplate ethical considerations, privacy concerns, and data security issues. Additional research is necessary to address these limitations and further the comprehension of digital health interventions in the realm of asthma management.

## 5. Conclusions

In summation, digital health interventions harbor the potential to bolster the management of asthma and associated psychological dimensions in children and adolescents. The studies examined in this article suggest that eHealth interventions, mixed reality tools, mHealth technology-enhanced nurse-guided interventions, and smartphone applications integrating EMA and Bluetooth-enabled sensors for asthma inhalers can significantly enhance symptom self-management, quality of life, and mental health outcomes. However, the domain of digital health interventions for the treatment of anxiety or depression in children and adolescents with long-term physical conditions remains limited, necessitating further research to ascertain their effectiveness and feasibility across diverse populations and settings. The formulation of tailored interventions that cater to the specific needs and preferences of patients with asthma and related psychological factors is vital for ensuring sustained and equitable utilization of these technologies. Future studies should concentrate on enhancing methodological rigor, generalizability, and long-term outcomes, as well as addressing concerns about accessibility, privacy, and data security. Fundamentally, digital health interventions have the potential to affect a significant difference in public health for individuals living with chronic conditions, including asthma. These interventions merit further investigation and development to improve patient outcomes and quality of life.

**Author Contributions:** P.C.: Conceptualization, Validation, Investigation, Supervision, Writing—original draft, Writing—review and editing; G.C.P.: Conceptualization, Investigation, Writing—original draft, Writing—review and editing; M.C.: Conceptualization, Investigation, Writing—original draft, Writing—review and editing; R.P.: Conceptualization, Investigation, Writing—original draft, Supervision, Writing—review and editing; M.C.Q.: Conceptualization, Investigation, Writing—

original draft, Writing—review and editing. All authors have read and agreed to the published version of the manuscript.

**Funding:** This research received no external funding.

**Institutional Review Board Statement:** Not applicable.

**Informed Consent Statement:** Not applicable.

**Conflicts of Interest:** The authors declare no conflicts of interest.

## Abbreviations

| | |
|---|---|
| AR | Augmented Reality |
| BA | Behavioral Activation |
| CBT | Cognitive Behavioral Therapy |
| DHIs | Digital Health Interventions |
| EMA | Ecological Momentary Assessment |
| ME | Motivational Enhancement |
| PTSS | Posttraumatic Stress Symptoms |
| VR | Virtual Reality |

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
