# Peer review of "Psychological Factors, Digital Health Technologies, and Best Asthma Management as Three Fundamental Components in Modern Care: A Narrative Review"

_applsci, doi:10.3390/app14083365_

Round 1
Reviewer 1 Report
Comments and Suggestions for Authors
Dear Authors.
Upon reviewing the manuscript, I have identified several areas where significant improvements are needed to enhance the overall quality of the paper.
Firstly, the search strategy employed in this study appears to be unconventional and lacks clarity. Please provide a clear rationale for your search strategy, including the selection criteria and databases used. Additionally, if certain databases such as EMBASE were not utilized, the authors should provide reasons for their exclusion.
Secondly, the structure of the results section lacks coherence. The presentation of findings should follow a logical sequence that allows readers to easily understand the flow of information.
Add a clear justification for the selection of included studies and demonstrate adherence to systematic review principles.
I have provided specific comments and recommendations in the PDF file that I will share with you.
Kind regards.

Author Response
Dear Reviewer,
We sincerely appreciate your thorough review of our manuscript and the insightful comments you provided. We are committed to addressing the areas for improvement you have identified to enhance the overall quality of our article.
1. Search Strategy: We apologize for any confusion regarding our search strategy. We utilized Summon, which encompasses a variety of databases including Elsevier, HeinOnLine, LexisNexis, IEEE, Jstor, ProQuest Health & Medical Complete, PsycArticles, Springer, Wiley, Web of Science, Pubmed, etc. It's important to clarify that our study is a Narrative review.
2. Eligibility criteria: We wish to acknowledge that we noticed some confusion in the eligibility criteria and have taken your valuable feedback into account to further improve the clarity and consistency of our inclusion and exclusion criteria. We have thoroughly reviewed the eligibility criteria and adjusted them based on your observations and the suggestions provided. This revision has allowed us to define the selection parameters more precisely.
3. Search filters: Given the limited availability of studies addressing our research question, we opted to employ multiple search strategies to ensure comprehensive coverage of the existing literature. Each search strategy was designed to capture different aspects or dimensions of the topic under investigation. By employing different search approaches, we sought to minimize the risk of overlooking relevant studies and to provide a more in-depth examination of the topic. While recognizing the potential availability of validated search filters, we found that existing filters were unable to adequately capture the breadth and specificity needed for the scope of our study. Therefore, we adapted our search strategies accordingly to ensure inclusiveness and comprehensiveness in identifying relevant literature. In any case, we decided, as you suggested, to eliminate some steps to exclude systematic reviews from our search that we initially included, and in doing so we sought to improve the robustness and reliability of our analysis.
4. Results Section: Thank you for bringing to our attention the inclusion of two systematic reviews among the articles considered for our review. We have removed them from the results section accordingly. We have made efforts to revise the results section to make it clearer and more reader-friendly. By restructuring the section, we aimed to present the findings in a more logical sequence, facilitating easier comprehension for the readers
5. Exclusion of Embase: While Embase was initially included in our search, we encountered difficulty in finding studies using our search strings. Therefore, no studies from Embase were included in our review.
6. RCTs reporting PROMs: We have carefully considered your comments regarding the limited availability of literature on this topic. After thorough exploration, we indeed encountered a scarcity of Randomized Controlled Trials (RCTs) reporting Patient-Reported Outcome Measures (PROMs) in the existing literature. Given the paucity of RCTs in this area, we made the decision to broaden our inclusion criteria to encompass studies beyond RCTs. By including non-RCTs, such as observational studies or qualitative research, we aimed to capture a more comprehensive understanding of the topic and the perspectives of patients. Furthermore, we acknowledge the importance of addressing these gaps in the literature. As such, we are committed to contributing to the field by initiating our own study on this subject matter. Our upcoming research endeavors will focus on conducting a rigorous investigation to fill these existing gaps and provide valuable insights into the use of PROMs in [specify the context or medical condition.
We genuinely appreciate your specific comments and recommendations, and we will carefully incorporate them into our revisions. Should you require any further clarification or have additional questions, please do not hesitate to reach out to us.
Reviewer 2 Report
Comments and Suggestions for Authors
Thank you for the opportunity to review this interesting systematic review. The paper is well written and structured and the only feedback I would have is minor.
- Systematic review is not mentioned in title or keywords, would encourage including it one of these so that your paper can be located easier later.
- On line 51-52, some clarity on where these definitions of eHealth and mHealth come from.
- For editorial team, Table 1 needs to be on one page for readability.
- Have a look over your introduction of acronyms. Some (e.g PICO) are not unpacked.
- Was there a reason for excluding papers <2015?
- In discussion, you should include a sentence on the digital divide.
- Mixed Reality is used in this paper to describe AR and VR - but this isn't really correct. There are a few definitions for MR (not one that is unanimously agreed upon). Suggest to either not use the term MR, or to define what the authors mean by MR in this paper.
Comments on the Quality of English LanguageVery well written paper.
Some sentence could be shortened and more clear.
Some acronyms not correctly introduced.
Author Response
Dear Reviewer,
We sincerely appreciate your thoughtful comments on our choice of topic and the significance of incorporating digital technologies into patient care.
- Title: thank you for your suggestion. We added "narrative Review" in the title and keywords
- Clarity on definitions of eHealth and mHealth: We appreciate your feedback. eHealth and mHealth definitions have been better specified.
- Table 1 readability: Thank you for reporting this point. We have updated the layout of Table 1 to ensure it is on one page, improving its readability.
- Introduction of acronyms: We checked the acronyms in the text.
- Exclusion of papers before 2015: Our decision to exclude pre-2015 articles was based on the need to focus on the most recent and relevant evidence.
- Discussion on the digital divide: Thank you for highlighting this point. We added a section dedicated to digital divide in the discussion, exploring how it can affect access to and use of digital healthcare technologies.
- Mixed Reality terminology: We reviewed the use of the term "Mixed Reality" in our article and provided a clear definition of the concept in relation to the context of our study.
Reviewer 3 Report
Comments and Suggestions for Authors
I congratulate the authors on their choice of topic. In this age of evolving digital technologies, I have no way of not using them in the treatment and care of patients, especially when they influence better diagnosis, improved test results and treatment and prognosis. E-health is becoming increasingly popular especially among the younger generation of patients and modern technologies are nothing strange or incomprehensible to this part of the population. The situation is slightly worse among the elderly population where fear and apprehension and stress about new technologies are certainly prevalent, but this should not be an obstacle to further research in this area, enabling the use of e-health in the broad sense of the word for the whole population rather than a select few. The work is a very good inspiration for finding more and better solutions and conducting research in this area.
Author Response
Dear Reviewer,
We sincerely appreciate your thoughtful comments on our choice of topic and the significance of incorporating digital technologies into patient care. Your acknowledgment of the increasing popularity of e-health, resonates deeply with our understanding of the evolving healthcare landscape. However, we acknowledge the challenges that the elderly population may face in embracing these technologies due to apprehension and fear. We wholeheartedly agree that such apprehension should not impede further research and development in the realm of e-health. Instead, it should serve as a catalyst for creating inclusive solutions that cater to the needs of all demographics. Bridging the digital divide and ensuring equitable access to e-health services for the entire population is a critical endeavor that warrants ongoing exploration and innovation. Your recognition of our work as a source of inspiration for future research is truly motivating. We are committed to continuing our efforts to contribute to the advancement of e-health solutions, with a focus on inclusivity and accessibility for all.